# Downregulation of Mitochondrial Fusion Protein Expression Affords Protection from Canonical Necroptosis in H9c2 Cardiomyoblasts

**DOI:** 10.3390/ijms25052905

**Published:** 2024-03-02

**Authors:** Yuki Toda, Sang-Bing Ong, Toshiyuki Yano, Atsushi Kuno, Hidemichi Kouzu, Tatsuya Sato, Wataru Ohwada, Yuki Tatekoshi, Toshifumi Ogawa, Masaki Shimizu, Masaya Tanno, Masato Furuhashi

**Affiliations:** 1Department of Cardiovascular, Renal and Metabolic Medicine, Sapporo Medical University School of Medicine, Sapporo 060-8543, Japan; yuuki.89er@gmail.com (Y.T.); kouzu@sapmed.ac.jp (H.K.); sato.tatsuya@sapmed.ac.jp (T.S.); w.ohwada9@gmail.com (W.O.); tannom@sapmed.ac.jp (M.T.); furuhasi@sapmed.ac.jp (M.F.); 2Department of Medicine and Therapeutics (MEDT), Faculty of Medicine, Chinese University of Hong Kong (CUHK), Hong Kong, China; sangbingong@cuhk.edu.hk; 3Centre for Cardiovascular Genomics and Medicine (CCGM), Lui Che Woo Institute of Innovative Medicine, Chinese University of Hong Kong (CUHK), Hong Kong, China; 4Hong Kong Children’s Hospital (HKCH), Hong Kong Hub of Paediatric Excellence (HK HOPE), Kowloon Bay, Hong Kong, China; 5Neural, Vascular, and Metabolic Biology Thematic Research Program, School of Biomedical Sciences (SBS), Chinese University of Hong Kong (CUHK), Hong Kong, China; 6Department of Pharmacology, Sapporo Medical University School of Medicine, Sapporo 060-8543, Japan; 7Department of Cell Physiology and Signal Transduction, Sapporo Medical University School of Medicine, Sapporo 060-8543, Japan

**Keywords:** necroptosis, cardiomyocyte, mitochondrial fusion, RIP1, TAK1

## Abstract

Necroptosis, a form of necrosis, and alterations in mitochondrial dynamics, a coordinated process of mitochondrial fission and fusion, have been implicated in the pathogenesis of cardiovascular diseases. This study aimed to determine the role of mitochondrial morphology in canonical necroptosis induced by a combination of TNFα and zVAD (TNF/zVAD) in H9c2 cells, rat cardiomyoblasts. Time-course analyses of mitochondrial morphology showed that mitochondria were initially shortened after the addition of TNF/zVAD and then their length was restored, and the proportion of cells with elongated mitochondria at 12 h was larger in TNF/zVAD-treated cells than in non-treated cells (16.3 ± 0.9% vs. 8.0 ± 1.2%). The knockdown of dynamin-related protein 1 (Drp1) and fission 1, fission promoters, and treatment with Mdivi-1, a Drp-1 inhibitor, had no effect on TNF/zVAD-induced necroptosis. In contrast, TNF/zVAD-induced necroptosis was attenuated by the knockdown of mitofusin 1/2 (Mfn1/2) and optic atrophy-1 (Opa1), proteins that are indispensable for mitochondrial fusion, and the attenuation of necroptosis was not canceled by treatment with Mdivi-1. The expression of TGFβ-activated kinase (TAK1), a negative regulator of RIP1 activity, was upregulated and the TNF/zVAD-induced RIP1-Ser166 phosphorylation, an index of RIP1 activity, was mitigated by the knockdown of Mfn1/2 or Opa1. Pharmacological TAK1 inhibition attenuated the protection afforded by Mfn1/2 and Opa1 knockdown. In conclusion, the inhibition of mitochondrial fusion increases TAK1 expression, leading to the attenuation of canonical necroptosis through the suppression of RIP1 activity.

## 1. Introduction

Necroptosis constitutes regulated cell death that has morphological characteristics of necrosis, and it is triggered by receptor-interacting protein (RIP)1-dependent and -independent RIP3 activation and is essential for host defense against a number of DNA and RNA viruses, including influenza A viruses [1,2,3,4,5]. Importantly, necroptosis serves as a fail-safe system to mediate death in cells that are devoid of the caspase 8-mediated extrinsic apoptotic pathway [3,6]. On the other hand, results of recent studies have shown that necroptosis is a part of the mechanisms of inflammatory bowel disease, autoimmune diseases, and ischemia/reperfusion injury [3,4,7]. RIP1 activation is a pivotal step leading to downstream RIP3 activation in the canonical pathway of necroptosis induced by tumor necrosis factor-α (TNFα) and interferon-γ, whereas viruses and dsDNA directly activate RIP3 [3]. There has been much debate about the mechanism by which activated RIP3 induces necroptosis, but mixed lineage kinase domain-like (MLKL), a pseudokinase that lacks two of the three conserved catalytic residues in its kinase-like domain, is currently thought to be a primary executor of necroptosis since the pharmacological suppression of MLKL activity via the inhibition of disulfide bond formation between monomeric MLKL, as well as the genetic deletion of MLKL, completely inhibits necroptosis regardless of whether necroptosis is triggered by RIP1 activation [8,9].

Studies in the past decade have unveiled the detailed molecular mechanisms of the process from RIP1 activation to the execution of necroptosis mediated by MLKL. MLKL that is phosphorylated by RIP3 has been shown to translocate from the cytosol to the plasma membrane, leading to necrosis through the perturbation of membrane integrity [3,10]. Thus, plasma membrane disruption mediated by MLKL activation is thought to be a necessary event for the execution of necroptosis. In addition, RIP3 and MLKL undergo nucleocytoplasmic shuttling, and the inhibition of nuclear MLKL translocation attenuates TNFα-induced necroptosis [11,12,13]. Furthermore, the activation of canonical necroptotic signaling induced by TNFα and caspase inhibition elicit translocation of the RIP3/MLKL complex to the endoplasmic reticulum-mitochondria interface [14], suggesting the possible involvement of mitochondria in canonical necroptosis. There are several lines of evidences suggesting the crosstalk of TNFα receptor signaling and alterations to mitochondria dynamics, a coordinated process of mitochondrial fission and fusion for regulating mitochondrial homeostasis and cell fate [15,16]. TNFα has been shown to induce an increase in the protein expression of dynamin-related protein 1 (Drp1), a large cytoplasmic GTPase that serves as a fission promoter, and Drp1 phosphorylation at Ser616, leading to mitochondrial fragmentation through promotion of the fission process in several cell lines, including the H9c2 rat cardiomyoblasts [17,18,19]. Conversely, treatment with TNFα induces mitochondrial fusion in neonatal mouse cardiomyocytes transfected with TNFα receptor 1 (TNFR1) through the TNFR2-mediated upregulation of optic atrophy 1 (Opa1) [20]. Thus, two TNFα receptors, TNFR1 and TNFR2, exert opposing effects on mitochondria dynamics. Therefore, changes in mitochondrial dynamics and its regulatory protein expression are likely to modulate TNFα-induced canonical necroptotic signaling.

In the present study, we used H9c2 cells, rat cardiomyoblasts, since our previous studies revealed that H9c2 cells possess a TNFα-induced canonical necroptotic program [9,13,21], and we examined the impact of TNFα-induced necroptotic signaling on mitochondria dynamics, if any, and whether the inhibition of fusion or fission modulates the extent of necroptosis.

## 2. Results

### 2.1. Time-Dependent Changes in Mitochondrial Morphology after the Induction of Necroptosis

The canonical necroptotic pathway was activated by TNFα and the caspase inhibitor zVAD (TNF/zVAD; TNFα, 50 ng/mL; zVAD, 20 μM), as we previously reported [13,21]. The release of LDH into the culture medium was increased in a time-dependent manner after the addition of TNF/zVAD (Figure 1A), and the difference in LDH release between TNF/zVAD-treated cells and vehicle-treated control cells reached statistical significance 1 h after the addition of TNF/zVAD (20.1 ± 2.9% in TNF/zVAD-treated cells vs. 12.1 ± 0.6% in vehicle-treated control cells).

Changes in mitochondrial morphology following the administration of TNF/zVAD were analyzed at different time points with a focus on morphologically living cells. After TNF/zVAD treatment, the percentage of cells containing predominantly elongated mitochondria was reduced, and the difference between the percentages of those cells before and after TNF/zVAD addition reached statistical significance 4 h after the addition of TNF/zVAD (8.0 ± 1.2% at 0 h vs. 4.3 ± 0.9% at 4 h, Figure 1B,C), suggesting the occurrence of mitochondrial fission induced by TNF/zVAD. This reduction in elongated mitochondria was subsequently reversed at 8 h (9.3 ± 1.8%, Figure 1B,C) and significantly at 12 h and 24 h (16.3 ± 0.9% & 23.3 ± 2.4%, respectively, Figure 1B,C).

### 2.2. Changes in Mitochondrial Fusion/Fission Proteins following the Addition of TNF/zVAD

As the proportion of cells with elongated mitochondria changes throughout the different time points after exposure to TNF/ZVAD, we wanted to determine whether the profile of mitochondrial-shaping proteins is altered in line with the change in mitochondrial morphology. Among the mitochondrial-fusion-regulatory proteins, treatment with TNF/zVAD significantly increased the levels of mitofusin 1 (Mfn1), an isoform of GTPase that is required for mitochondrial outer membrane fusion, at 4 h and 12 h after treatment by 1.67-and 1.94-fold, respectively, and also increased the level of mitofusin 2 (Mfn2) at 1 h after treatment by 1.27-fold (Figure 2), whereas levels of Opa1, a mitochondrial inner membrane fusogen, remain unchanged. The mitochondrial fission regulatory proteins Drp1 and Fission 1 (Fis1) were not affected by TNF/zVAD treatment (Figure 2). In agreement with our recent study [13], a modest increase in the MLKL level was found after TNF/zVAD treatment (Figure 2).

### 2.3. Role of Mitochondrial Fusion/Fission Proteins in Necroptosis

To demonstrate the mechanistic link between TNF/zVAD-induced mitochondrial fission and necroptosis, the effects of the siRNA-mediated knockdown of mitochondrial fission proteins on TNF/zVAD-induced cell death were examined. siRNA-mediated Drp1 knockdown was confirmed based on Western blotting (Figure 3B) and the appearance of large mitochondria with elongated and enlarged shapes (Figure 3C), and Drp1 knockdown had no effects on TNF/zVAD-induced LDH release (Figure 3D). The experiments with Fis1 knockdown yielded similar results (Figure 4). The presence of different concentrations of mdivi-1, a small molecule inhibitor of Drp1, throughout TNF/zVAD treatment also did not cause any significant change to LDH release (Appendix A).

Next, the role of mitochondrial fusion proteins in necroptosis was examined. Mfn1 and Mfn2 contain homologous functional domains and complementarily work to maintain proper mitochondrial function mainly through the control of mitochondrial dynamics, though distinct roles of each isoform in mitochondrial function have also been reported [22]. In light of the purpose of this experiment, the double knockdown of Mfn1 and Mfn2 (Mfn1/2) was selected for subsequent experiments. The knockdown of Mfn1/2 successfully induced mitochondrial fragmentation (Figure 5C) and significantly reduced TNF/zVAD-induced LDH release (Figure 5D). The significant attenuation of TNF/zVAD-induced LDH release was also recapitulated by the knockdown of Opa1 (Figure 6). These protective effects of Mfn1/2 or Opa1 knockdown on TNF/zVAD-induced cell death were not reversed by treatment with Mdivi-1 (Appendix A). Collectively, the results indicate that the knockdown of mitochondrial fusion proteins is protective for TNF/zVAD-induced necroptosis possibly through a mitochondrial fission-independent mechanism.

### 2.4. Effects of Knockdown of Mitochondrial Fusion Proteins on Necroptotic Signaling

The mechanisms by which the knockdown of Mfn1/2 and Opa1 affords protection from TNF/zVAD-induced necroptosis were examined. TNF/zVAD induced the phosphorylation of RIP1 at Ser166 (Figure 7), a residue that positively regulates RIP1 activity, as shown in our previous study [9]. Both Mfn1/2 knockdown and Opa1 knockdown reduced TNF/zVAD-induced RIP1 phosphorylation, but the suppression of RIP3 expression was found only with Mfn1/2 knockdown (Figure 7), indicating that the alteration in the pathway from TNF receptor activation to RIP1 phosphorylation is a primary mechanism of the suppression of necroptosis mediated by the knockdown of mitochondrial fusion proteins.

We also investigated whether the enhanced activity of endogenous necroptosis suppressors upstream of RIP1 is involved in the protective mechanism of the knockdown of mitochondrial fusion proteins. Consistent with our earlier findings [9], TNF/zVAD-induced LDH release was aggravated by the addition of 5z7—a TAK1 inhibitor (Figure 8)—and it tended to be exaggerated by the addition of TPCA-1—an IKKα/β inhibitor—and BV6—an inhibitor of cellular inhibitor of apoptosis proteins 1 (Figure 8 and Appendix A). The addition of 5z7, but not the addition of TPCA-1 or BV6, partly canceled the Mfn1/2 knockdown-mediated protection from TNF/zVAD-induced cell death (Figure 8 and Appendix A). Similar results were obtained when 5z7 was administered in experiments with Opa1 knockdown (Figure 8). The expression of TAK1, a negative regulator of RIP1 activity, was similarly upregulated after the knockdown of Mfn1/2 and Opa1 (Figure 9), indicating that increased expression of TAK1 mediated by the knockdown of mitochondrial fusion proteins attenuates TNF/zVAD-induced necroptotic signaling. The knockdown of Mfn1/2 and Opa1 did not affect the TAK1 mRNA level (Figure 9), suggesting that post-translational mechanisms are responsible for the enhancement of TAK1 protein expression. As mitochondrial biogenesis has been associated with the prevention of mitochondrial dysfunction and regulation of necroptosis, we also assessed the level of PGC-1α as a predictor of mitochondrial biogenesis. In line with the increase in the proportion of cells with predominantly fragmented mitochondria, we also observed an increase in the PGC-1α level in cells with Mfn1/2 knockdown. However, we detected no changes in PGC-1α expression following Opa1 knockdown (Appendix A).

## 3. Discussion

The findings from this study are as follows: (i) in line with LDH release, TNF/zVAD also induces an increase in the proportion of cells with predominantly elongated mitochondria probably due to the increased level of Mfn1; (ii) siRNA-mediated knockdown of the mitochondrial fusion-regulating proteins reduces TNF/zVAD-induced LDH release; (iii) Mfn1/2 knockdown reduces RIP1-Ser166 phosphorylation and increases levels of TAK1 (Figure 10).

Crosstalk between mitochondrial fission and regulated cell death has been reported. Drp1-dependent mitochondrial fragmentation is thought to be associated with outer membrane permeabilization, which precedes caspase-3 activation for the execution of apoptosis in a wide range of organisms, including mammals [23]. Although cell death induced by ischemia/reperfusion (hypoxia/reoxygenation), primarily mediated by mitochondria permeability transition pore (mPTP) opening, was prevented by the pharmacological short-term inhibition of fission mediated by Mdivi-1 in cultured cardiomyocytes and in vivo animal models [24,25], Drp1 ablation also provoked mPTP opening, leading to necrosis in cardiomyocytes and mouse embryonic fibroblasts (MEFs) [26,27]. Conflicting results also exist regarding the relationship between fission and necroptosis [28]. Results of a pioneering study by Wang Z et al. showed that Drp1 dephosphorylation at Ser637, an inhibitory phosphorylation site, mediated by the mitochondrial protein phosphatase PGAM5, leading to increased mitochondrial fission, plays a pivotal role in the execution of necroptosis, as indicated by results showing that Drp1 knockdown attenuates necroptosis in HeLa and HT-29 cells, i.e., human cancer cells [29]. However, the protective effect of Drp1 knockdown on necroptosis was not recapitulated in subsequent studies in which L929 cells, mouse fibroblast cells, and MEFs were used [30,31]. In the present study, TNFα-induced canonical necroptotic signaling induced mitochondrial fragmentation together with a slight enhancement of Drp1-Ser616 phosphorylation in H9c2 cardiomyoblasts at the time when a significant increase in LDH release in the culture medium was found (Figure 1 and Figure 2). However, the pharmacological inhibition of Drp1 activity mediated by Mdivi-1, as well as genetic manipulation via Drp1 or Fis1 knockdown, had no effects on TNF/zVAD-induced LDH release. These conflicting results are not easily reconciled, but differences in cell types, e.g., human cell line vs. rodent-derived cells, may be an explanation for the conflicting results, as shown by the species differences in the molecular mechanisms of RIP3-mediated MLKL activation [32,33]. Nevertheless, the involvement of mitochondria in the process from RIP1 activation to the execution of necroptosis has been questioned based on the results of vigorous analyses, including a study in which cells that were nearly devoid of mitochondria underwent necroptosis [28,34].

A salient finding in the present study is the favorable effect of the downregulation of mitochondrial fusion protein expression on necroptosis. Although Mfn isoforms have different functions, the loss of each Mfn isoform does not necessarily show an obvious phenotype since the Mfn isoforms complementarily work to maintain proper mitochondrial function [22,35]. In addition, a study by Kawalec et al. showed that expression levels of mitochondria biogenesis-regulating genes, including PGC-1α, were increased in Mfn2-null MEFs in comparison to the levels in wild-type MEFs, leading to the preservation of mitochondrial respiration and mitochondrial DNA content [36]. In contrast, the combined deletion of Mfn1/2 led to severe functional defects with mitochondrial dysfunction. Indeed, a conditional Mfn1/2 deletion in mouse hearts and MEFs induced mitochondrial fragmentation evoked by unopposed fission together with the dissipation of mitochondrial membrane potential, respiratory impairment, and mitochondrial ROS production, but evidence of apoptosis and necrosis was not found in cardiomyocytes of conditional Mfn1/2-deletion mice [26], suggesting the existence of compensatory mechanisms to maintain cell survival. In the present study, the upregulation of PGC-1α protein expression was also found in H9c2 cells transfected with Mfn1/2 siRNA, but this was not the case for cells transfected with Opa1 siRNA. In contrast, the knockdown of Mfn1/2 and knockdown of Opa1 similarly reduced RIP1 activation as shown by reduced RIP1-Ser166 phosphorylation and increased protein expression levels of TAK1, a protein that functions as a molecular switch to activate the TNFα-induced NFκB pro-survival pathway together with the suppression of necroptosis in cardiomyocytes and MEFs [37]. The results of an earlier study by Li et al. showed that TNFR1 activation induces the association of TAK1 with RIP1 and activates the IκB kinase-NFκB pro-survival pathway [37]. In the condition in which TAK1 activity is inhibited, RIP1 dissociates from TAK1, leading to the binding of RIP1 to caspase-8 and the Fas-associated death domain, which triggers the necroptotic pathway through RIP1 activation [37]. These earlier findings in cardiomyocytes and MEFs were supported by the results of the present study: pharmacological TAK1 inhibition mitigated the protection afforded by the knockdown of Mfn1/2 and Opa1. Importantly, TAK1 has been shown to play pivotal roles in the preservation of mitochondrial function: increased mitochondrial reactive oxygen species and mitochondrial dysfunction, leading to skeletal muscle atrophy, in inducible skeletal muscle-specific TAK1 knockout mice were reported [38,39]. Thus, the upregulation of TAK1 expression mediated by the knockdown of mitochondrial fusion-regulating proteins seems to serve as a compensatory pathway for protecting cells from necroptosis and maintaining mitochondrial function. Although mitochondria seem to be dispensable for the execution of necroptosis [28,34], alterations in necroptotic signaling mediated by disturbances of mitochondrial function may play a role in the pathophysiology of various diseases in which mitochondrial dysfunction is involved. Further detailed investigations are needed to demonstrate whether the modulation of mitochondrial morphology and the mitochondrial fusion proteins at specific time points also changes canonical necroptotic signaling in vivo.

There are several limitations in the present study. First, the exact role of the shift in mitochondrial morphology from a predominantly fragmented phenotype during initial TNF/zVAD treatment to a predominantly elongated phenotype following prolonged TNF/zVAD exposure remains unknown. Whether promoting mitochondrial fission reverses necroptotic signaling and LDH release remains to be investigated. In addition, the quantitative determination of mitochondrial size/length by using unbiased approaches, such as machine learning, are needed to confirm the results of this study. Second, although this study focused on the association between necroptosis and mitochondria dynamics, the treatment with TNF/zVAD may also modulate mitochondrial function through alterations in mitochondrial dynamics, independent of its effects on necroptosis. Third, the mechanism by which downregulation of the expression of mitochondrial fusion proteins contributes to an increase in TAK1 expression remains unclear. Since the knockdown of Mfn1/2 and Opa1 had no effect on TAK1 mRNA levels, post-transcriptional modifications and protein-protein interactions may play a role. Fourth, considering the partial block of protection afforded by Mfn1/2 and Opa1 knockdown mediated by TAK1 inhibition, further detailed investigations are needed to demonstrate the underlying mechanism: Mfn2 ablation-induced upregulation of the glycolytic pathway and mTORC2-Akt signaling are possible candidates [40,41,42,43]. Fifth, there are conflicting results about the significance of TAK1 in the cardiomyocytes subjected to pathological stress: TAK1 activation is a promoter of pressure overload-induced cardiac dysfunction, whereas TAK1 ablation is detrimental for a pressure-overloaded heart through the enhancement of necroptotic signaling [37,44,45]. Thus, the significance of increased TAK1 expression mediated by the ablation of mitochondrial fusion-regulating proteins in cardiomyocytes is likely to be context-dependent. Finally, further studies are needed to determine whether the results of the present study can be extrapolated to adult cardiomyocytes, though H9c2 cells have similarities to neonatal and adult cardiomyocytes [9].

## 4. Materials and Methods

### 4.1. Chemical Compounds

TNFα, Mdivi-1, TPCA-1, and 5z7 from Sigma Aldrich (St. Louis, MO, USA), Z-Val-Ala-DL-Asp-fluoromethylketone (zVAD) from Promega (Madison, WI, USA), and BV6 from ApexBio Technology (Houston, TX, USA) were used. siRNA was purchased from Dharmacon (Lafayette, CO, USA).

### 4.2. Cell Culture and Transfection

H9c2 cells (American Type Culture Collection, Manassas, VA, USA) were cultured in DMEM (4.5 g/L glucose) supplemented with 10% fetal bovine serum and antibiotics. The knockdown of Drp1, Fis1, Mfn1, Mfn2, and Opa1 was performed based on the transfection of siRNA against rat Drp1 (M-088074-01-0005), rat Fis1 (M-080846-01-0005), rat Mfn1 (M-099253-01-0005), rat Mfn2 (M-094723-01-0005), and rat Opa1 (M-086996-01-0005) using Lipofectamine^TM^ RNAiMAX (Thermo Fisher Scientific, Waltham, MA, USA) according to the manufacturer’s protocol.

### 4.3. Experimental Protocols and Cell Death Assay

H9c2 cells were treated with a combination of 50 ng/mL TNFα and 20 μM zVAD (TNF/zVAD) or a vehicle through the addition of them to the culture medium [9,13,21]. Inhibitors, i.e., 5z7 (50 nM), TPCA1 (0.10~0.25 μM), BV6 (0.3~1.0 μM), and Mdivi-1 (10~50 μM), were added to the culture medium at the same time that the cells received TNF/zVAD or a vehicle. The siRNAs were transfected into H9c2 cells 24 h before the addition of TNF/zVAD. The extent of LDH release from cells, a marker of necrosis, was determined by measuring the activity of lactate dehydrogenase (LDH) in the culture medium and LDH activity after freeze-thawing of the cells at the end of experiments according to the manufacturer’s protocol (CytoTox 96 Non-Radioactive Cytotoxicity assay kit, Promega, Madison, WI, USA). The percentage of LDH activity in the culture medium to LDH activity after freeze-thawing of the cells, i.e., total cellular LDH activity, served as an index of TNF/zVAD-induced necroptosis, as we previously reported [9,13,21].

### 4.4. Western Blotting

Whole cell lysates were obtained through homogenization in ice-cold CHAPS buffer containing 20 mM HEPES (pH 7.5), 120 mM NaCl, 1 mM EDTA, 50 mM NaF, 0.3% CHAPS, 0.5 mM Na_3_VO_4_, and protease/protease inhibitor cocktails. Western blotting was performed as previously reported [9,13,21]. Antibodies used were as follows: phospho-Drp1 (Ser616), phospho-Drp1 (Ser637), and Drp1, phospho-RIP1 (Ser166), and RIP1, RIP3, and TAK1 (1:1000, Cell Signaling Technology, Beverly, MA, USA); Mfn1/2 and Mfn2 (1:1000, Abcam, Cambridge, UK); MLKL and PGC1α (1:1000, Merck Millipore, Damstadt, Germany); Fis1 (1:1000, Gene Tex, Irvine, CA, USA); Opa1 (1:1000, BD Biosciences, San Jose, CA, USA); and vinculin (1:5000, Sigma Aldrich, St Louis, MO, USA).

### 4.5. Fluorescence Microscopy Analysis

Mitochondrial morphology was assessed as previously reported with modifications [24]. Cells, after the treatment or transfection with siRNAs, were stained with 1 μM MitoTracker Red (Invitrogen, Waltham, MA, USA) for 15 min to stain mitochondria. Nuclear staining with Hoechst 33342 (Invitrogen, Waltham, MA, USA) was then performed. Eighty randomly chosen cells per group were designated as either containing predominantly (>50%) elongated or predominantly (>50%) fragmented mitochondria, as we previously reported [24].

### 4.6. mRNA Quantification

The isolation of total RNA from cells was performed by using an RNeasy Mini Kit (Qiagen, Valencia, CA, USA), followed by the synthesis of first-strand cDNA by using a SuperScript VILO^TM^ cDNA synthesis kit (Thermo Fisher Scientific, Waltham, MA, USA). DNA amplification was performed in a StepOne^TM^ system (Thermo Fisher Scientific, Waltham, MA, USA) by using Go Taq qPCR Master Mix (Promega, Madison, WI, USA) and oligonucleotide primers for rat TAK1 (Rn01437012_m1, Thermo Fisher Scientific, Waltham, MA, USA) and rat β-actin (Rn00667869, Thermo Fisher Scientific, Waltham, MA, USA).

### 4.7. Statistical Analysis

Data are presented as the means ± standard error of the mean. Results were analyzed using an unpaired *t*-test for comparisons between two groups. One-way analysis of variance (ANOVA) was used to detect significant differences when more than 2 groups were present. When ANOVA indicated a significant overall difference, multiple comparisons of the groups were performed by performing a Tukey post hoc test. A difference was considered to be statistically significant when the *p*-value was less than 0.05. All of the statistical analyses were performed with JMP Pro15 software (SAS Institute, Cary, NC, USA).

## 5. Conclusions

Activation of the canonical necroptotic pathway is associated with a change in mitochondrial dynamics, starting with an initial fragmentation followed by subsequent elongation. The Mdivi-1-mediated pharmacological inhibition of Drp1 activity, as well as genetic manipulation via Drp1 or Fis1 knockdown, has no effects on the extent of canonical necroptosis. Conversely, the genetic downregulation of mitochondrial fusion protein expression affords protection from canonical necroptosis through a reduction in RIP1-Ser166 phosphorylation mediated by enhanced TAK1 expression in H9c2 cardiomyoblasts.

## Figures and Tables

**Figure 1 ijms-25-02905-f001:**
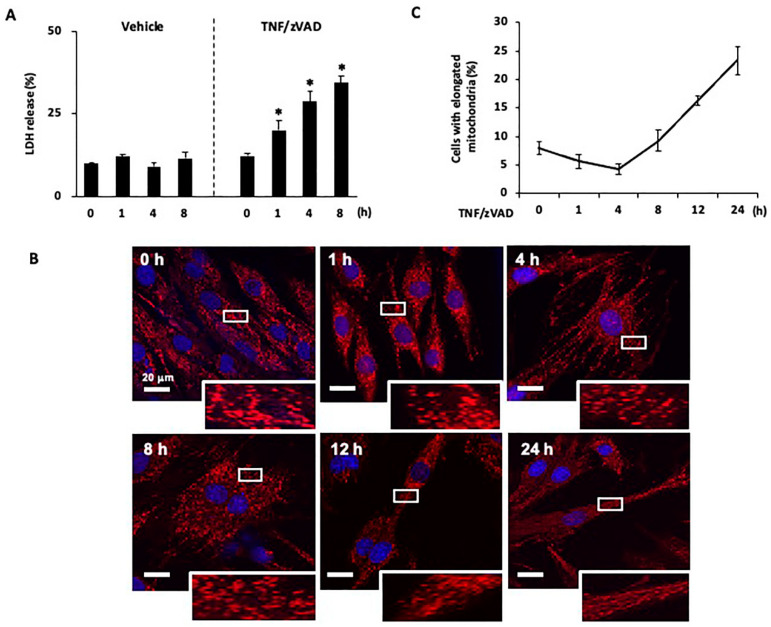
Time-dependent changes in mitochondrial morphology after the induction of necroptosis. (**A**) Time course of necroptosis induced by treatment with TNFα and zVAD (TNF/zVAD; TNFα, 50 ng/mL; zVAD, 20 μM) in H9c2 cells. Culture media were collected at 4, 8, and 12 h after the addition of TNF/zVAD, and the extent of necroptosis was determined by quantifying lactate dehydrogenase (LDH) in the culture media. * *p* < 0.05 vs. baseline (0 h). *n* = 4 in each group. (**B**) Representative confocal images of H9c2 cells stained with MitoTracker Red (red) and Hoechst 33342 (blue). Images were acquired at baseline and 0, 4, 8, 12, and 24 h after the addition of TNF/zVAD. (**C**) Percentage of H9c2 cells containing predominantly (>50%) elongated mitochondria over a 24 h period post-TNF/zVAD treatment. *n* = 4 experiments for each time point.

**Figure 2 ijms-25-02905-f002:**
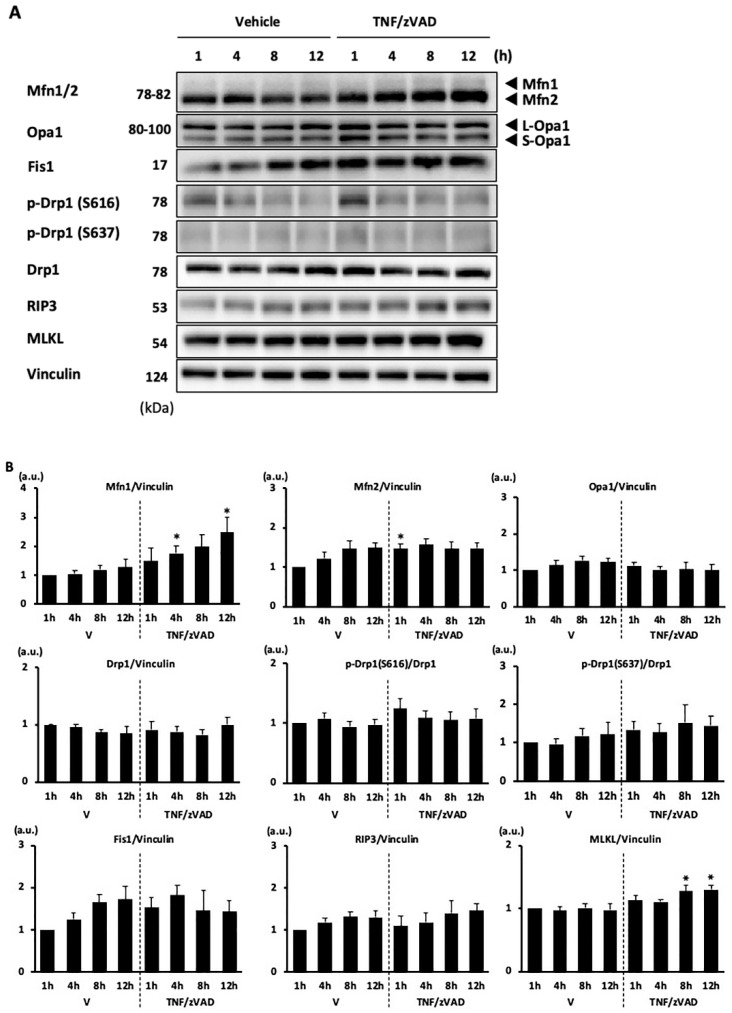
Changes in mitochondrial fusion/fission proteins following the addition of TNF/zVAD. (**A**,**B**) Representative Western blots (**A**) and results of densitometric analyses (**B**) are shown. H9c2 cells were treated with a vehicle (V) or TNFα and zVAD (TNF/zVAD; TNFα, 50 ng/mL; zVAD, 20 μM). Mfn = mitofusin, Opa1 = optic atrophy 1, Fis1 = fission 1, Drp1 = dynamin-related protein 1, RIP3 = receptor-interacting protein kinase 3, MLKL = mixed lineage kinase domain-like protein. Vinculin was used as a loading control. *n* = 6 in each group. * *p* < 0.05 vs. vehicle at the same time point.

**Figure 3 ijms-25-02905-f003:**
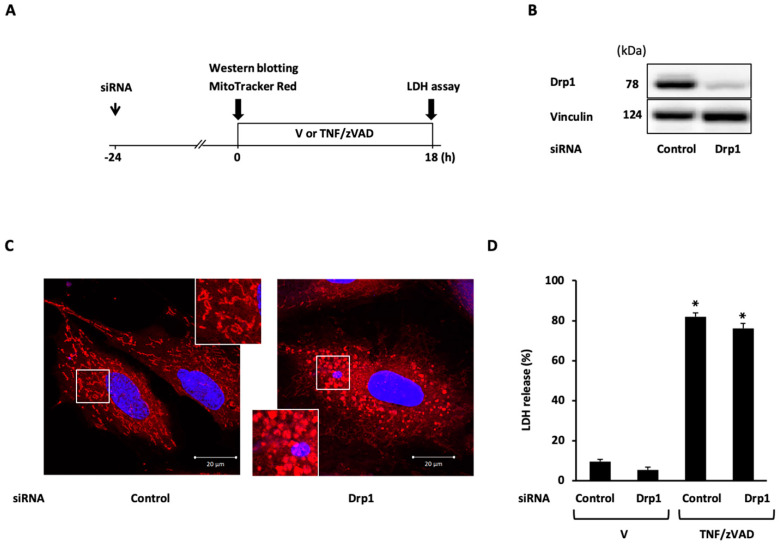
Effect of dynamin-related protein 1 (Drp1) knockdown on TNF/zVAD-induced LDH release. (**A**) Experimental protocol. Control siRNA and Drp1 siRNA were transfected into H9c2 cells 24 h before the addition of a vehicle (V) or TNFα and zVAD (TNF/zVAD; TNFα, 50 ng/mL; zVAD, 20 μM). (**B**) Representative images of Western blots in H9c2 cells transfected with control or Drp1 siRNA. Vinculin was used as a loading control. (**C**) Representative confocal images of H9c2 cells stained with MitoTracker Red (red) and Hoechst 33342 (blue). (**D**) Effects of Drp1 knockdown on TNF/zVAD-induced LDH release. LDH release from the cells was determined at 18 h after the addition of V or TNF/zVAD. *n* = 7 in each group. * *p* < 0.05 vs. V-treated cells transfected with control siRNA.

**Figure 4 ijms-25-02905-f004:**
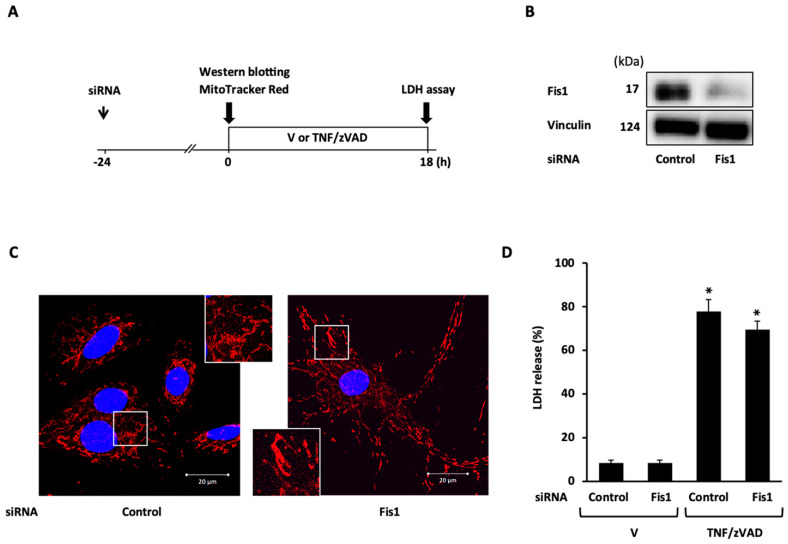
Effect of fission 1 (Fis1) knockdown on TNF/zVAD-induced LDH release. (**A**) Experimental protocol. Control siRNA and Fis1 siRNA were transfected into H9c2 cells 24 h before the addition of a vehicle (V) or TNFα and zVAD (TNF/zVAD; TNFα, 50 ng/mL; zVAD, 20 μM). (**B**) Representative images of Western blots in H9c2 cells transfected with control or Fis1 siRNA. Vinculin was used as a loading control. (**C**) Representative confocal images of H9c2 cells stained with MitoTracker Red (red) and Hoechst 33342 (blue). (**D**) Effects of Fis1 knockdown on TNF/zVAD-induced LDH release. LDH release from the cells was determined at 18 h after the addition of V or TNF/zVAD. *n* = 7 in each group. * *p* < 0.05 vs. V-treated cells transfected with control siRNA.

**Figure 5 ijms-25-02905-f005:**
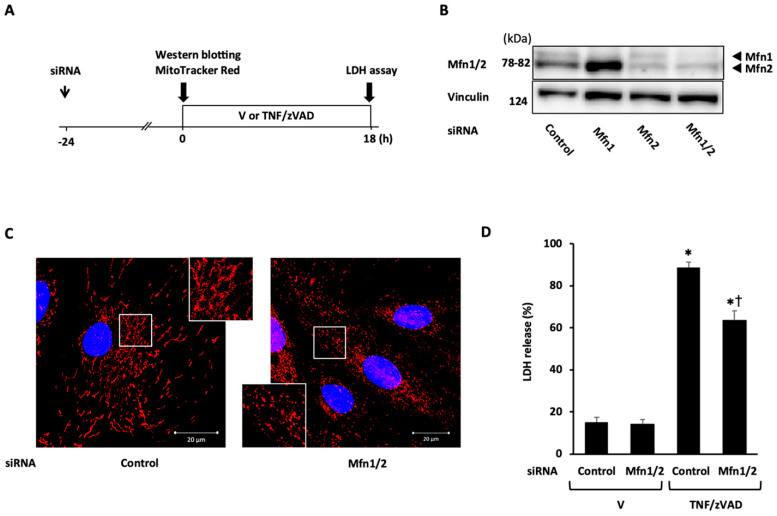
Effect of mitofusin 1/2 (Mfn1/2) knockdown on TNF/zVAD-induced LDH release. (**A**) Experimental protocol. Control siRNA and Mfn siRNAs were transfected into H9c2 cells 24 h before the addition of a vehicle (V) or TNFα and zVAD (TNF/zVAD; TNFα, 50 ng/mL; zVAD, 20 μM). (**B**) Representative images of Western blots in H9c2 cells transfected with control siRNA or Mfn siRNAs. Vinculin was used as a loading control. (**C**) Representative confocal images of H9c2 cells stained with MitoTracker Red (red) and Hoechst 33342 (blue). (**D**) Effects of Mfn1/2 knockdown on TNF/zVAD-induced LDH release. LDH release from the cells was determined at 18 h after the addition of V or TNF/zVAD. *n*= 6 in each group. * *p* < 0.05 vs. V-treated cells transfected with control siRNA. † *p* < 0.05 vs. TNF/zVAD-treated cells transfected with control siRNA.

**Figure 6 ijms-25-02905-f006:**
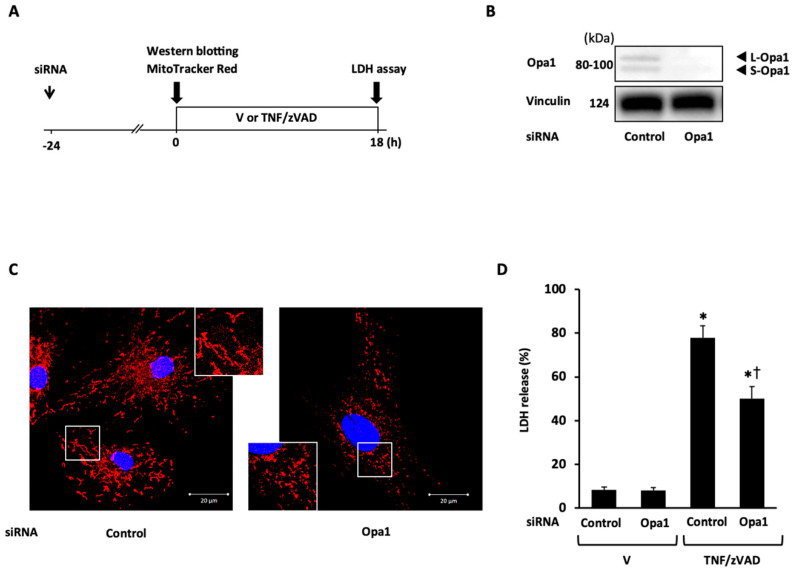
Effect of optic atrophy-1 (Opa1) knockdown on TNF/zVAD-induced LDH release. (**A**) Experimental protocol. Control siRNA and Opa1 siRNA were transfected into H9c2 cells 24 h before the addition of a vehicle (V) or TNFα and zVAD (TNF/zVAD; TNFα, 50 ng/mL; zVAD, 20 μM). (**B**) Representative images of Western blots in H9c2 cells transfected with control or Opa1 siRNA. Vinculin was used as a loading control. (**C**) Representative confocal images of H9c2 cells stained with MitoTracker Red (red) and Hoechst 33342 (blue). (**D**) Effects of Opa1 knockdown on TNF/zVAD-induced LDH release. LDH release from the cells was determined at 18 h after the addition of V or TNF/zVAD. *n* = 7 in each group. * *p* < 0.05 vs. V-treated cells transfected with control siRNA. † *p* < 0.05 vs. TNF/zVAD-treated cells transfected with control siRNA.

**Figure 7 ijms-25-02905-f007:**
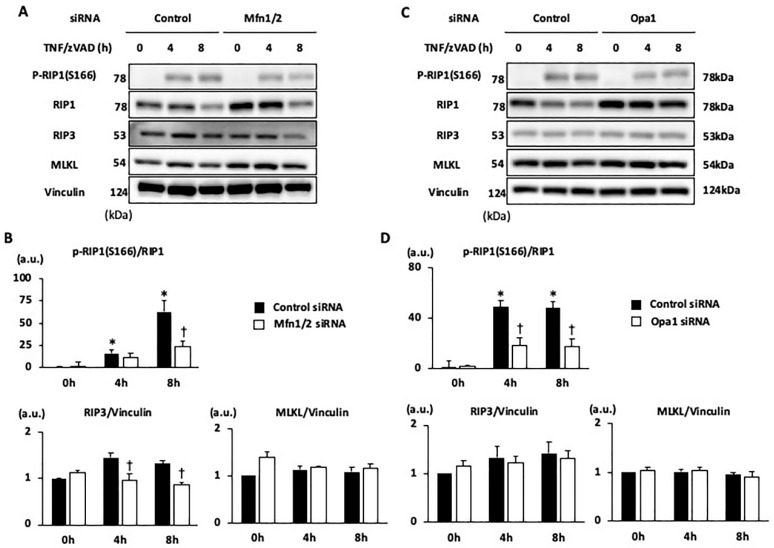
Effect of mitochondrial fusion-regulating protein knockdown on TNF/zVAD-induced necroptotic signaling. (**A**,**C**) Representative Western blots for proteins regulating necroptotic pathways are shown. The siRNAs were transfected into H9c2 cells 24 h before the addition of a vehicle (V) or TNFα and zVAD (TNF/zVAD; TNFα, 50 ng/mL; zVAD, 20 μM). Mfn = mitofusin, Opa1 = optic atrophy-1, RIP1 = receptor-interacting protein kinase 1, RIP3 = receptor-interacting protein kinase 1, MLKL = mixed lineage kinase domain-like protein. Vinculin was used as a loading control. (**B**,**D**) Results of densitometric analyses are shown. *n* = 6 in each group. * *p* < 0.05 vs. cells transfected with control siRNA (0 h). † *p* < 0.05 vs. TNF/zVAD-treated cells transfected with control siRNA at the same time point.

**Figure 8 ijms-25-02905-f008:**
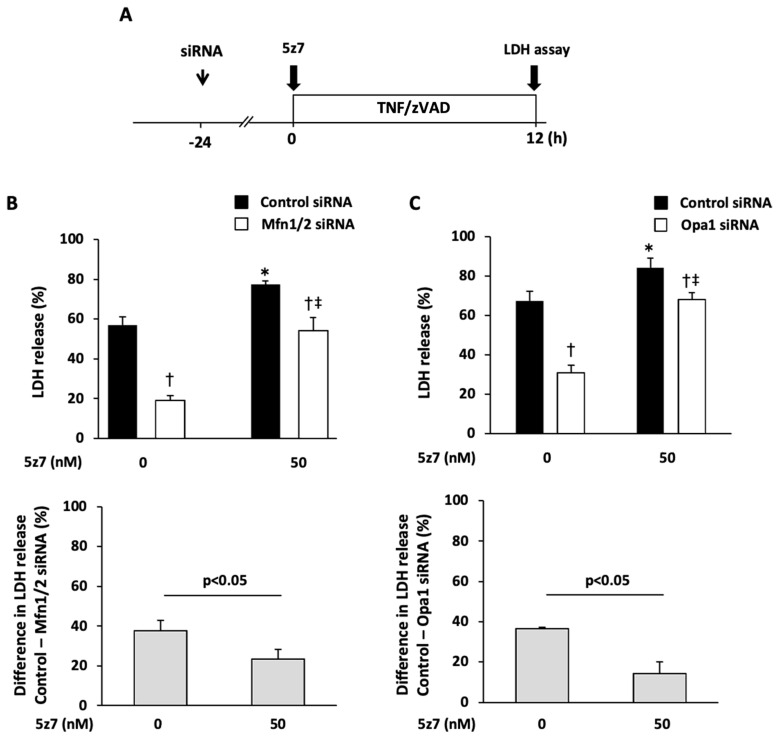
Effects of inhibitors of endogenous necroptosis suppressors upstream of RIP1 on the protection afforded by the knockdown of mitochondrial fusion-regulating protein. (**A**,**B**) Effects of 5z7, a TGFβ-activated kinase (TAK1) inhibitor, on protection from TNF/zVAD-induced cell death mediated by the knockdown of Mfn1/2 and Opa1. The experimental protocol (**A**) and results of quantitative analyses (**B**,**C**) are shown. The siRNAs were transfected into H9c2 cells 24 h before the addition of TNFα and zVAD (TNF/zVAD; TNFα, 50 ng/mL; zVAD, 20 μM). Mfn = mitofusin, Opa1 = optic atrophy-1. *n* = 8 in each group. * *p* < 0.05 vs. cells transfected with control siRNA (0 h). † *p* < 0.05 vs. cells transfected with control siRNA and treated with similar concentrations of inhibitors. ‡ *p* < 0.05 vs. cells transfected with Mfn1/2 or Opa1 siRNA (0 h).

**Figure 9 ijms-25-02905-f009:**
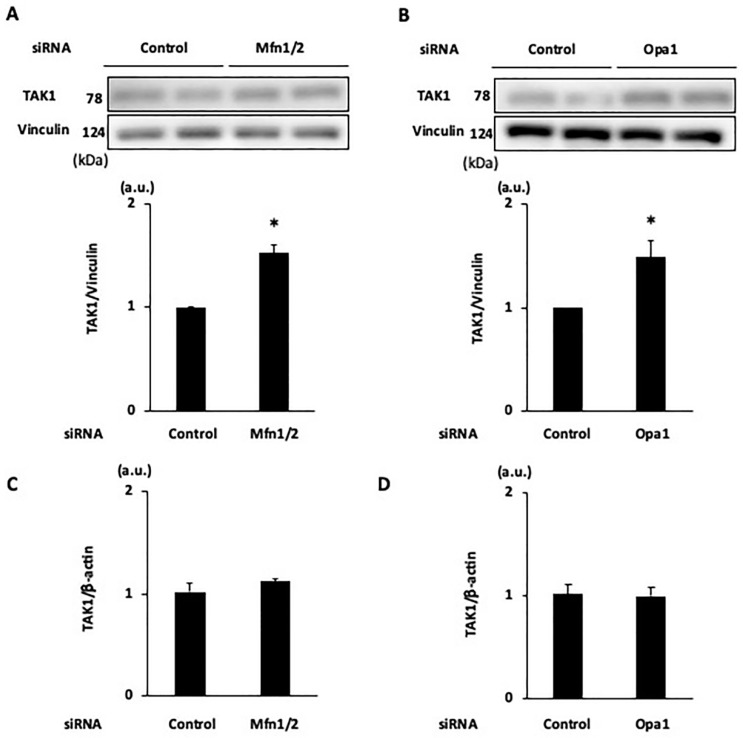
Effects of the knockdown of mitochondrial fusion-regulating protein on TAK1 expression. (**A**,**B**) Effects of the knockdown of Mfn1/2 (**A**) and Opa1 (**B**) on TAK1 protein levels. Representative Western blots and results of densitometric analyses are shown. The siRNAs were transfected into H9c2 cells, and cell lysates were collected 24 h after the transfection of siRNAs. Mfn = mitofusin, Opa1 = optic atrophy-1, TAK1 = transforming growth factor-β-activated kinase 1. Vinculin was used as a loading control. *n* = 4 in each group. * *p* < 0.05 vs. cells transfected with control siRNA. (**C**,**D**) Effects of the knockdown of Mfn1/2 (**C**) and Opa1 (**D**) on TAK1 mRNA levels.

**Figure 10 ijms-25-02905-f010:**
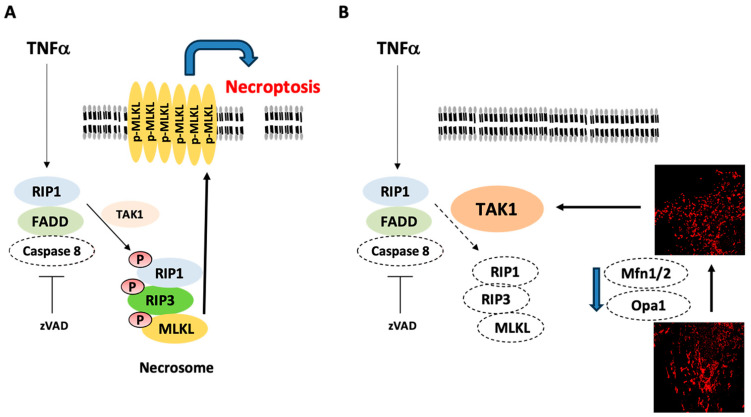
Proposed mechanism by which the downregulation of mitochondrial fusion protein expression affords protection from canonical necroptosis. (**A**) TNFα-induced canonical necroptotic pathway. (**B**) The downregulation of mitochondrial fusion protein expression affords protection from canonical necroptosis through a reduction in RIP1-Ser166 phosphorylation mediated by enhanced TAK1 expression. RIP1 = receptor-interacting protein kinase 1, FADD = Fas-associated death domain, TAK1 = transforming growth factor-b-activated kinase 1, RIP3 = receptor-interacting protein kinase 3, MLKL = mixed lineage kinase domain-like protein, Mfn = mitofusin, Opa1 = optic atrophy 1.

## Data Availability

Data are contained within the article.

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
