# Peer review of "Downregulation of Mitochondrial Fusion Protein Expression Affords Protection from Canonical Necroptosis in H9c2 Cardiomyoblasts"

_ijms, 2024, doi:10.3390/ijms25052905_

Round 1

Reviewer 1 Report

Comments and Suggestions for Authors

The manuscript by Toda et al. reports on important new findings regarding mitochondrial morphology in necroptosis process. The study has been performed in H9c2 cardiomyoblast cell line. The authors demonstrated that downregulation of mitochondrial fusion protein expression protected from canonical necroptosis. Mitochondrial fusion increased TAK1 expression, that attenuated canonical necroptosis via suppression of RIP1 activity.

The findings will be of importance for our understanding and pharmacological regulation of cell necroptosis. The manuscript is well concepted and written. The methods are appropriate.

I have several minor comments:

Concentration of the compounds 5z7, TPCA1, BV6 and Mdivi-1 (line 374) is missing, please add.

Concentration of antibodies used in Western Blot is also missing, please add.

3.    Line 379-381: ” The percentage of LDH activity in the culture medium to LDH activity after freeze-thawing of the cells served as an index of TNF/zVAD-induced necrosis, i.e., necroptosis”. This method should be explained and the reference is necessary.

4.    It would be important to provide the readers a schema with explanation of the main findings in context of necroptotic pathways.

Reviewer 2 Report

Comments and Suggestions for Authors

The manuscript “Downregulation of mitochondrial fusion protein expression affords protection from canonical necroptosis in H9c2 cardiomyoblasts” by Toda Y., et al explores the involvement of fusion and fission proteins in the induction of cell death in the model of TNFalpha/zVAD-induced necroptosis.

The manuscript seems to be valuable for the understanding of mechanisms of cardiac cell death. Nevertheless, some aspects deserve further clarification and explanation.

1.       Quantitative determination of mitochondrial elongation is problematic: in fact, authors use two poor defined parameters “elongated mitochondria” (what does it mean? What length is sufficient to name mitochondria “elongated”?) and number of cells which contain >50% elongated mitochondria. Will it be better to use integral parameter, for instance, average length of mitochondria?

2.       Dynamics of the level of fusion/fission proteins (Fig. 2). Since in vehicle control some changes in protein levels may be observed, it would be interesting to see zero points, especially for Mfn1/Mfn2 and Fis1. How author could explain the fact, that RIP3, a major mediator of necroptosis involved in the plasma membrane perturbations and LDH release remains on the same level, while unrelated to necroptosis Fis1 rises considerably? According to bars, Drp1 level is unchanged, while according to blots, its level decreases at least twice. Is this figure representative?

3.        According to description of Figs 3C and 4C, knockdown of Drp1 and Fis1 should cause mitochondrial elongation. However, it is not obvious from the pictures presented. Could Authors provide some statistical data? Similarly, in description to Fig 5C, KD of Mfn1/2 induces mitochondrial fragmentation, however the picture is not convincing. In Fig. 6C the brightness of mitochondria is so low that it is almost impossible to assess their shape even with maximal magnification (though mito seem smaller).

4.       It is important to know, how TNFalpha/zVAD treatment and Drp1-, Fis1-, Mfn1/2-, and Opa1-KD effect the mitochondrial functionality. Do mitochondria preserve membrane potential and respiration rates?

5.       The scale on the ordinate axis in Figs. 3D, 4D, and 6D is not convenient. Is it possible to add more points?

6.       In Figs. 3-6, the LDH assay was performed at 18-h after TNFalpha/zVAD treatment. Why data on the p-RIP1, RIP1, RIP3, and MLKL levels (Fig. 7) cover the 8-h period only?

To conclude, the presented manuscript seems interesting and valuable, while several issues should be addressed to remove all doubts about the correctness of data interpretation.

Round 2

Reviewer 2 Report

Comments and Suggestions for Authors

I am satisfied with Author’s responses. Some shortcomings are still preserved in the manuscript but the Authors acknowledge these shortcomings and discuss them in the appropriate section.